# Experience of gender-based violence and its effect on depressive symptoms among Indian adolescent girls: Evidence from UDAYA survey

Ratna Patel[1], Samriddhi S. Gupte[2], Shobhit Srivastava[3], Pradeep Kumar[3], Shekhar Chauhan[4]*, Mani Deep Govindu[5], Preeti Dhillon[6]

1 Department of Public Health and Mortality Studies, International Institute for Population Sciences, Mumbai, India, 2 International Institute for Population Sciences, Mumbai, India, 3 Department of Mathematical Demography & Statistics, International Institute for Population Sciences, Mumbai, India, 4 Department of Population Policies and Programmes, International Institute for Population Sciences, Mumbai, India, 5 Karnataka Health Promotion Trust (KHPT), Bengaluru, Karnataka, India, 6 Department of Mathematical Demography & Statistics, International Institute for Population Sciences, Mumbai, India

* shekhariips2486@gmail.com

**Data Availability Statement:** https://dataverse.harvard.edu/dataset.xhtml?persistentId=doi:10.7910/DVN/RRXQNT.

## Abstract

### Background

Violence against women has been one of the most prominent issue and a major public health concern. It is a principle violation of basic human rights and has both physical and mental effect on the victim. This study focuses on married and unmarried girls aged 15 to 19 years, belonging to Uttar Pradesh and Bihar, India. This study attempts to examine depression level among married and unmarried girls who have faced violence against them. With the unprecedented growth in social networking, online digital platform and its accessibility, the study also brings out the pertinent aspect of internet based violence and its psychological outcome on adolescent girls. Hence, the study can be seen as an important and needed value addition to the existing pool of knowledge on the subject.

### Methods

The study uses Understanding the lives of adolescents and young adults (UDAYA) project data for Uttar Pradesh and Bihar. Depressive symptoms among adolescent girl is the outcome variable of the study. Descriptive statistic and bivariate analysis has been used to get to preliminary results. Chi-squared test is used to test the significant of variables. Further, multi-variate analysis (logistic regression) was used.

### Results

Almost 29, 23 and 26 percent of married adolescent girls had faced emotional, physical and sexual violence respectively. It was found that about five per cent of unmarried and eight per cent of married girls had high depressive symptoms. It was found that unmarried adolescent girls who had witnessed their father beating mother were 71 percent more likely to suffer from higher depressive symptoms [OR: 1.71, 1.09–2.69]. Adolescents who faced

**Funding:** This paper was written using data collected as part of Population Council's UDAYA study, which is funded by the Bill and Melinda Gates Foundation and the David and Lucile Packard Foundation. No additional funds were received for the preparation of the paper.

**Competing interests:** The authors have declared that no competing interests exist.

perpetrated bullying had 90 per cent [OR: 1.90, 1.32–2.72] and 86 per cent [OR: 1.86, 0.98–3.52] higher likelihood to suffer from higher depressive symptoms.

## Conclusion

The study goes beyond intimate partner violence and includes various covariates to explain the association between violence and depressive symptoms among married and unmarried adolescents. Hence, more inclusive policies are needed to address the issue of violence against women as the spectrum of the violence is expanding with time.

## Introduction

World Health Organization defines violence as "The intentional use of physical force or power, threatened or actual that results in or has a high likelihood of resulting in injury, death, psychological harm, mal-development or deprivation." The inclusion of word "power" broadens "the conventional understanding of violence include those acts that result from a power relationship, including threats and intimidation" [1]. Violence against women has long been a prominent and widespread area of concern in public health. It is also a principle violation of basic human rights, which impairs, particularly, women's right to life, right to freedom from torture and other cruel, inhuman or degrading treatments or punishments, and to the highest attainable standard of physical and mental health [2–4].

Violence against women take various forms ranging from emotional to sexual. Global studies show women all over the world face violence in various forms at hands of various people [5,14,15]. The most common forms of violence that a woman faces are domestic violence, abuse and sexual violence [5]. Worldwide 35 percent of the women have experienced intimate partner or non-partner sexual violence [6]. The 2013–2014 Crime Survey of England showed that around 2.2 percent (approximately 355000) of women aged 16 to 59 have suffered some kind of sexual assault along with 8.5 percent women (1.5 million) had experienced domestic abuse in the past year [7]. In United States, more than 10 million adults experience domestic violence annually [8]. Also, the number of intimate partner victims has seen an increase by 42 percent from 2016 to 2018 in U.S [9]. In India, 30 percent of women have experienced physical violence since the age of 15 years and around 4 percent of ever pregnant women have experienced physical violence during pregnancy. According to National Family Health Survey, 2015–16, of the total ever married women, 33 percent have experienced spousal violence in various forms such as physical, sexual or emotional. Of this, the most common form of violence is physical violence, followed by emotional violence [10]. In India, the problem of gender based violence has taken a grave turn. With a number of policies and acts to strengthen position of women within the society, the crimes against women are still seeing new peaks. However, there does exist a regional disparity [11]. According to NCRB Crime in India 2018 statistic, highest number of dowry related deaths occurred in Uttar Pradesh, followed by Bihar (252.1 and 111.1 deaths per million female population, respectively) [12]. Uttar Pradesh and Bihar have also reported the highest number of cases of kidnapping or abduction of girls below 18 years to compel her marriage (444.9, 294.2 women per million women respectively), whereas, Assam, West Bengal and Rajasthan have reported the highest number of cases in the section of cruelty by husband or relatives (crime rate of 66.7 percent, 35.0 percent, 33.0 percent respectively) [12].

Given the diversity and disparity that exists in India, it becomes imperative to look into the factors that affect violence against women. Broadly these factors can be clustered as those

falling under *Extrinsic factors* and those under *Intrinsic factors* [13]. However, the underlying cause of violence of any kind remains to be the power imbalance [14]. In many cultures, including India, males dominate and control the power hierarchies and thus, uses several forms of violence over women to keep their supremacy intact [15]. Using domestic violence against women is a stereotypical notion by man to maintain or regain his masculinity [16]. Intrinsic factors are the factors pertaining to individual or personal characteristics such as age, education, income etc. Some factors that do influence the acceptance of violence are also witnessing violence as a child or having faced violence at a young age [17]. Age is also seen as a critical factor, 15 to 19, that is adolescence, is seen to be the most vulnerable age where almost all women face or witness some form of violence [18].

Witnessing or facing violence has long term effect on the mental health of women. Literature asserts that domestic violence and abuse increases the likeliness of depressive disorder [19]. Not just, physical violence but psychological domestic violence and abuse has also shown to have similar detrimental effect as physical violence with respect to depressive disorder, PTSD (Post Traumatic Stress Disorder) and psychological stress [20–22]. However, there is dearth of work on association between violence and depressive symptoms among married and unmarried adolescents in India. Although, adolescents have been identified as at risk age group, yet almost no study delves into their psychological response to having witnessed or experienced violence.

The current study attempts to bridge the gap that exists in the literature on the association between violence witnessed/experienced by adolescents and depressive symptoms among them. The study focuses on married and unmarried girls aged 15 to 19, belonging to Uttar Pradesh and Bihar, India. With the unprecedented growth in social networking, online digital platform and its accessibility, the study also brings out the pertinent aspect of internet based violence and its psychological outcome on adolescent girls. Hence, the study can be seen as an important and needed value addition to the existing pool of knowledge on the subject. This study hypothesize that adolescents experiencing violence would show higher depressive symptoms.

## Methods

### Data

The present study carried out data from Understanding the lives of adolescents and young adults (UDAYA) project survey, which was conducted in two Indian states Uttar Pradesh and Bihar, in 2016 by Population Council under the guidance of Ministry of Health and Family Welfare, Government of India. With the use of a multi-stage sampling procedure, the survey gathered information on family, media, community environment, and quality of transitions to young adulthood indicators, and provide the estimates for states as a whole as well as rural and urban of both states. The sample size for Uttar Pradesh and Bihar was 10,161 and 10,433 adolescents aged 10–19 years, respectively. For this study, a total sample of 12,972 adolescent girls aged 15–19 years was considered. Among them, 7,766 respondents were unmarried, 313 were married, but no Gauna was performed and 4,893 were currently married at the time of the survey. We dropped the 313 married, but no Gauna performed cases as the data did not have information about these respondents for physical violence by a parent during the last 12 months. In some parts of India, girls participate in a marriage ceremony at a very young age, but they do not begin a sexual relationship or cohabitate with their husbands until a ceremony known as *gauna* has taken place [23]. *Gauna* is most common in North India where parents may wait several years between the marriage and *gauna* until the girl attains puberty or is deemed mature enough to begin living with her husband [23]. The effective sample size for this study was 12,599 adolescent girls aged 15–19 years after adjusting for missing data from

some married and unmarried adolescent girls. The sample was representative for Uttar Pradesh and Bihar to make the estimates representative and to account for the multi stage systematic sampling design, we used survey weights across the analysis [24,25]. Informed consent was sought from each individual to be interviewed, among unmarried adolescents in ages 10–17, consent was also sought from a parent or guardian. Additionally, names were never recorded in the computer form in which data were collected. In order to preserve the confidentiality of the respondent or the parent/guardian, signing the consent form was optional; however, the interviewer was required to sign a statement that she or he had explained the content of the consent form to the respondent or parent.

This study uses data which is secondary in nature and therefore does not require any ethical approval from any institutional review board. The data collection for UDAYA survey was approved by Population Council, New Delhi and ethical review board of Population Council, New Delhi approved the questionnaire that was used in the field work.

**Outcome variable.** Depressive symptom among adolescent girls was the outcome variable of this study. The respondent was asked about the symptoms for past two weeks only. The questions included, a. had trouble falling asleep or sleeping too much, b. feeling tired or having little energy, c. poor appetite or eating too much, d. trouble concentrating on things, e. had little interest or pleasure in doing things f. feeling down, depressed or hopeless, g. feeling bad about yourself, h. been moving or speaking slowly, i. had thoughts that respondent would be better off dead. All the above questions were asked on a scale of four i.e., 0 "not at all", 1 "less than once a week", 2 "one week or more" and 3 "nearly every day". The scale of 27 points was then generated using egen command in STATA 14. The variable was treated as count variable for the analytical purpose (Cronbach alpha: 0.86). A score of 0–9 was considered as mild/minimal depressive symptoms whereas the score of 10–27 was considered as moderate to severe category [24–27]. The variable was recoded as a binary variable for analytical purpose i.e. 0 as low which include mild/minimal symptoms and 1 as high which include moderate to severe symptoms.

**Exposure variables.** The variables included gender based violence among unmarried and married adolescent girls. The following categorization was done for the analytical purpose.

*Gender based violence among unmarried adolescent girls*

1. Parental violence was coded as 1 means 'Yes' depicting if the respondent witnessed father ever beating mother in the last 12 months and 0 means 'No,' otherwise.

2. Physical violence was recorded as 1 'Yes' if the respondent experienced physical violence by a parent during the last 12 months and 0 'No'; otherwise. This question was asked to unmarried girls only.

3. Sexual violence was coded as 'Yes' Ever experienced forced sex (attempted or forced) and 'No'; otherwise.

*Intimate partner violence among married adolescent girls*

1. Separate questions were asked to married girls only regarding emotional, physical, and sexual violence. Emotional violence was defined as if the husband humiliate respondent in front of others coded as 1 'Yes' and 0 'No,' otherwise.

2. Physical violence was recorded as 1 'Yes' if the husband ever slapped, twisted or pulled hair, pushed/shook or throe something, kicked dragged beaten, burnt on purpose, attacked with a knife to the respondent and 0 'No,' otherwise.

3. Sexual violence was defined as 'Yes' if the husband ever forced the respondent to have sex in the last 12 months and 'No'; otherwise.

Justification over wife-beating was recoded from the question 'is it right for a husband to beat his wife' coded 'Justified and 'not justified.' Perpetrated bullying in the last 12 months (ever teased/beaten a girl/boy/weaker for any reason in the last 12 months) were coded as 'Yes' and 'No.' Perpetrated bullying means that they were carrying out bullying. If the respondent experienced mobile or internet-based harassment, it was recorded as 'Yes' and 'No' if the respondent did not experience this kind of harassment.

Age of the adolescent girls grouped into two categories: early adolescent (15–17 years) and late adolescent (18–19 years). Education was categorized as: no education, 1–7 years of schooling, 8–9 years of schooling, and 10 & above years of schooling. Place of residence was given in the survey as rural and urban. Caste was grouped as Scheduled Caste/Scheduled Tribe (SC/ST) and Other Backward Class and Others. The Scheduled Caste include "untouchables"; a group of population which is socially segregated and financially/economically by their low status as per Hindu caste hierarchy. The Scheduled Castes (SCs) and Scheduled Tribes (STs) are among the most disadvantaged socio-economic groups in India. The OBC are the group of people who were identified as "educationally, economically and socially backward". The OBC's are considered low in traditional caste hierarchy but are not considered as untouchables. The "other" caste category are identified of having higher social status [28].

Religion was categorized into two groups: Hindu and non-Hindu. The survey measured household economic status, using a wealth index composed of household asset data on ownership of selected durable goods, including means of transportation, as well as data on access to a number of amenities. The wealth index was constructed by allocating the following scores to a households reported assets or amenities. Index scores had ranged from 0 to 57. Households were ranked according to the index score (divided into five groups) based on five quintiles using xtile function in STATA 14. The wealth quintile was coded as poorest, poorer, middle, richer and richest. Access to the mobile phone was coded as 'Yes' if the respondent had their mobile phone or have access to a family member's mobile phone and 'No' otherwise. Similarly, access to the internet was coded as 'Yes' if the respondent had access to the internet on mobile or computer and 'No'; otherwise. Exposure to mass media was coded as "Yes" if the respondent had daily or weekly exposure to television, film, radio, or media, and "No" otherwise. The data was available for two states Uttar Pradesh and Bihar.

**Statistical analysis.**   The bivariate and multivariate analysis adopted to fulfil the study objective. First, univariate (sample distribution) analysis was done to observe the frequency distribution of adolescent girls. Second, bivariate analysis was used to find the association between outcome and independent variables. A chi-square test used to test the level of significance between variables. Finally, logistic regression was used to estimates the effect of different exposure variables on high depressive symptoms among adolescent girls. The results were presented in the form of odds and 95% confidence interval (CI).

$$ln\left(\frac{p}{1-p}\right) = \alpha + \beta_1 X_1 + \beta_2 X_2 + \beta_3 X_3 \ldots \ldots \beta_n X_n$$

Where $\beta_0, \ldots, \beta_M$ are regression coefficient indicating the relative effect of a particular explanatory variable on the outcome. These coefficients change as per the context in the analysis in the study.

## Results

### Exposure to gender-based Violence and characteristics of adolescent girls

Table 1 depicts the socio-demographic characteristics of adolescent girls aged 15–19 years and their exposure to gender-based violence (GBV). It was found that about five per cent of

**Table 1. Characteristics of adolescent girls aged 15–19 years and their exposure to GBV.**

| Background Characteristics | Unmarried | | Married | |
|---|---|---|---|---|
| | Percentage | N | Percentage | N |
| **Depressive symptoms** | | | | |
| Low | 95.1 | 7,383 | 92.1 | 4,452 |
| High | 4.9 | 383 | 7.9 | 380 |
| **Witnessed father ever beating mother** | | | | |
| No | 92.4 | 7,174 | – | |
| Yes | 7.6 | 592 | | |
| **Experienced physical violence by a parent** | | | | |
| No | 90.7 | 7,042 | – | |
| Yes | 9.3 | 724 | | |
| **Experienced Sexual Violence** | | | | |
| No | 97.6 | 7,582 | – | |
| Yes | 2.4 | 184 | | |
| **Experienced Emotional Violence** | | | | |
| No | | | 71.4 | 3,451 |
| Yes | | | 28.6 | 1,382 |
| **Experienced Physical Violence** | | | | |
| No | | | 77.1 | 3,724 |
| Yes | | | 22.9 | 1,109 |
| **Experienced Sexual Violence** | | | | |
| No | | | 73.9 | 3,572 |
| Yes | | | 26.1 | 1,261 |
| **Justification over wife-beating** | | | | |
| Justified | 19.8 | 1,539 | 26.7 | 1,290 |
| Not justified | 80.2 | 6,227 | 73.3 | 3,543 |
| **Experienced mobile phone/internet-based harassment** | | | | |
| No | 95.5 | 7,414 | 93 | 4,493 |
| Yes | 4.5 | 352 | 7.0 | 340 |
| **Perpetrated bullying** | | | | |
| No | 85.1 | 6,610 | 97.2 | 4,697 |
| Yes | 14.9 | 1,156 | 2.8 | 136 |
| **Age groups (in years)** | | | | |
| 15–17 | 71.9 | 5,580 | 26.6 | 1,285 |
| 18–19 | 28.2 | 2,186 | 73.4 | 3,547 |
| **Education level (in years)** | | | | |
| No education | 7.6 | 593 | 27.3 | 1,318 |
| 1–7 | 18.8 | 1,459 | 23.5 | 1,135 |
| 8–9 | 33.3 | 2,585 | 25.0 | 1,210 |
| 10 & above | 40.3 | 3,130 | 24.2 | 1,170 |
| **Place of residence** | | | | |
| Urban | 17.3 | 1,344 | 14.8 | 713 |
| Rural | 82.7 | 6,422 | 85.2 | 4,120 |
| **Caste** | | | | |
| SC/ST | 22.1 | 1,715 | 28.6 | 1,384 |
| OBC | 57.2 | 4,444 | 60.7 | 2,935 |
| Others | 20.7 | 1,607 | 10.6 | 514 |
| **Religion** | | | | |

*(Continued)*

**Table 1.** (Continued)

| Background Characteristics | Unmarried | | Married | |
|---|---|---|---|---|
| | Percentage | N | Percentage | N |
| Hindu | 77.5 | 6,019 | 84.1 | 4,065 |
| Non-Hindu | 22.5 | 1,747 | 15.9 | 768 |
| **Wealth quintile** | | | | |
| Poorest | 11.9 | 923 | 13.6 | 656 |
| Poorer | 17.1 | 1,327 | 19.6 | 947 |
| Middle | 20.9 | 1,626 | 23.4 | 1,133 |
| Richer | 25.2 | 1,955 | 25.5 | 1,233 |
| Richest | 24.9 | 1,934 | 17.9 | 865 |
| **Mobile access** | | | | |
| No mobile | 12.1 | 941 | 4.9 | 235 |
| Own Mobile | 8.4 | 650 | 35.6 | 1,719 |
| Family member's | 79.5 | 6,175 | 59.6 | 2,879 |
| **Internet access** | | | | |
| No | 91.4 | 7,094 | 96.2 | 4,648 |
| Yes | 8.7 | 672 | 3.8 | 185 |
| **Mass media exposure** | | | | |
| No | 38.4 | 2,984 | 52.4 | 2,534 |
| Yes | 61.6 | 4,782 | 47.6 | 2,298 |
| **State** | | | | |
| Uttar Pradesh | 55.9 | 4,338 | 35.5 | 1,713 |
| Bihar | 44.1 | 3,428 | 64.6 | 3,120 |
| **Total** | 100.0 | 7,766 | 100.0 | 4,833 |

*SC/ST*: *Scheduled caste/scheduled tribe*; *GBV*: Gender-based violence; *OBC*: *Other backward Class*; *Violence among married women come under intimate partner violence; depressive symptoms (low include scores from 0–9 and high include scores 10–27).*

unmarried and eight percent of married girls had high depressive symptoms. Nearly eight percent of unmarried adolescents witnessed their father beat their mother. Around nine percent of unmarried girls experienced physical violence by their parents, and two percent had experienced sexual violence. About 29 percent, 23 percent and 26 percent of married adolescent girls experienced emotional, physical and sexual violence, respectively.

About 20 percent of unmarried and 27 percent of married girls justified wife beating by husband. Moreover, around five percent of unmarried and seven percent of married girls had experienced mobile phone/internet-based harassment. Perpetrated bullying was more common among unmarried adolescent girls (15%) than married girls (3%). Only eight percent of unmarried and 36 percent of married girls had access over their own mobile phones. However, only nine percent of unmarried and 4 percent of married girls had internet access.

## Prevalence of high depressive symptoms among adolescents by exposure to GBV and their characteristics

Percentage of adolescent girls who reported high depressive symptoms by gender-based violence and other background characteristics, according to their marital status is presented in *Table 2*. The prevalence of high depressive symptoms was higher among unmarried girls who witnessed mother beaten by their father (9%). Moreover, unmarried adolescent girls who faced physical violence by their parents (8%) and if ever experienced sexual violence (11%)

**Table 2. Percentage of adolescent girls who reported high depressive symptoms by gender-based violence and other background characteristics, according to their marital status.**

| Background Characteristics | Unmarried | | Married | |
|---|---|---|---|---|
| | High depressive symptoms | p-value | High depressive symptoms | p-value |
| **Witnessed father ever beating mother** | | *** | | |
| No | 4.6 | | | |
| Yes | 8.9 | | | |
| **Experienced physical violence by a parent** | | *** | | |
| No | 4.6 | | | |
| Yes | 8.4 | | | |
| **Experienced Sexual Violence** | | *** | | |
| No | 4.8 | | | |
| Yes | 11.1 | | | |
| **Experienced Emotional Violence** | | | | *** |
| No | | | 5.7 | |
| Yes | | | 13.4 | |
| **Experienced Physical Violence** | | | | *** |
| No | | | 6.2 | |
| Yes | | | 13.5 | |
| **Experienced Sexual Violence** | | | | *** |
| No | | | 6.9 | |
| Yes | | | 10.7 | |
| **Justification over wife-beating** | | | | * |
| Justified | 5.5 | | 6.2 | |
| Not justified | 4.8 | | 8.5 | |
| **Experienced mobile phone/internet-based harassment** | | *** | | ** |
| No | 4.7 | | 7.5 | |
| Yes | 10.2 | | 12.6 | |
| **Perpetrated bullying** | | *** | | ** |
| No | 4.3 | | 7.7 | |
| Yes | 8.3 | | 14.8 | |
| **Age groups (In years)** | | *** | | |
| 15–17 | 4.3 | | 7.9 | |
| 18–19 | 6.6 | | 7.9 | |
| **Education level (In years)** | | * | | |
| No education | 6.7 | | 7.8 | |
| 1–7 | 4.0 | | 7.3 | |
| 8–9 | 5.7 | | 9.3 | |
| 10 & above | 4.4 | | 7.1 | |
| **Place of residence** | | * | | |
| Urban | 6 | | 8.8 | |
| Rural | 4.7 | | 7.7 | |
| **Caste** | | ** | | |
| SC/ST | 6.4 | | 7.9 | |
| OBC | 4.2 | | 7.9 | |
| Others | 5.3 | | 7.9 | |
| **Religion** | | | | *** |
| Hindu | 4.7 | | 7.3 | |
| Non-Hindu | 5.7 | | 10.9 | |

*(Continued)*

**Table 2.** (Continued)

| Background Characteristics | Unmarried | | Married | |
|---|---|---|---|---|
| | High depressive symptoms | p-value | High depressive symptoms | p-value |
| **Wealth quintile** | | * | | |
| Poorest | 5.3 | | 7.1 | |
| Poorer | 4.1 | | 8.9 | |
| Middle | 3.5 | | 8.6 | |
| Richer | 6.0 | | 6.7 | |
| Richest | 5.4 | | 8.1 | |
| **Mobile access** | | ** | | ** |
| No mobile | 4.7 | | 5.3 | |
| Own | 7.8 | | 9.5 | |
| Family member's | 4.7 | | 7.1 | |
| **Internet access** | | | | ** |
| No | 4.8 | | 7.6 | |
| Yes | 5.9 | | 14.3 | |
| **Mass media exposure** | | * | | |
| No | 4.1 | | 7.9 | |
| Yes | 5.4 | | 7.9 | |
| **State** | | | | * |
| Uttar Pradesh | 4.9 | | 9.3 | |
| Bihar | 5.0 | | 7.1 | |

***p<0.001

**p<0.05

*p<0.10

SC/ST: Scheduled caste/scheduled tribe; OBC: Other Backward Class; Violence among married women come under intimate partner violence; depressive symptoms (low include scores from 0–9 and high include scores 10–27).

had higher prevalence of depressive symptoms. Moreover, the prevalence of depressive symptoms was higher among married girls who experienced emotional violence (13%), physical violence (14%), and sexual violence (11%). Adolescent girls who experienced harassment over mobile phone/internet had higher prevalence of depressive symptoms (unmarried-10% and married-13%). Interestingly, depressive symptoms was higher among adolescent girls who perpetrated bullying (unmarried-8% and married-15%) i.e. If the respondent bullied someone, she had higher depressive symptoms. Similarly, depressive symptoms was more prevalent among adolescent girls who had own mobile phone access (unmarried-8% and married-10%). Married adolescent girls who had internet access had significantly higher prevalence of depressive symptoms (14%).

## Association between exposure to GBV and depressive symptoms among adolescent girls

Table 3 provides the estimates from binary logistic regression analysis for high depressive symptoms among adolescent girls by background characteristics. It was found that unmarried adolescents who witnessed their father ever beating their mother had 71 percent higher likelihood to suffer from higher depressive symptoms than their counterpart [OR: 1.71, 1.09–2.69]. Similarly, unmarried adolescents who experienced physical violence by parents had 62 per cent higher likelihood to suffer from higher depressive symptoms than who did not experience [OR: 1.62, 1.16–2.25]. Unmarried girls who experienced sexual violence had 98 percent higher

**Table 3. Estimates from binary logistic regression analysis for high depressive symptoms among adolescent girls by background characteristics.**

| Background Characteristics | Unmarried | Married |
|---|---|---|
| | OR [95% CI] | OR [95% CI] |
| **Witnessed father ever beating mother** | | |
| No (**Ref**) | | |
| Yes | 1.71 [1.09–2.69]** | |
| **Experienced physical violence by a parent** | | |
| No (**Ref**) | | |
| Yes | 1.62 [1.16–2.25]*** | |
| **Experienced Sexual Violence** | | |
| No (**Ref**) | | |
| Yes | 1.98 [0.94–4.19]* | |
| **Experienced Emotional Violence** | | |
| No (**Ref**) | | |
| Yes | | 2.10 [1.42–3.11]*** |
| **Experienced Physical Violence** | | |
| No (**Ref**) | | |
| Yes | | 1.63 [1.13–2.35]*** |
| **Experienced Sexual Violence** | | |
| No (**Ref**) | | |
| Yes | | 1.11 [0.81–1.53] |
| **Justification over wife beating** | | |
| Justified (**Ref**) | | |
| Not justified | 0.89 [0.59–1.34] | 1.59 [1.09–2.33]** |
| Experienced mobile phone/internet based harassment | | |
| No (**Ref**) | | |
| Yes | 1.78 [1.02–3.11]** | 1.37 [0.8–2.37] |
| **Perpetrated bullying** | | |
| No (**Ref**) | | |
| Yes | 1.90 [1.32–2.72]*** | 1.86 [0.98–3.52]* |
| **Age groups (In years)** | | |
| 15–17 (**Ref**) | | |
| 18–19 | 1.81 [1.34–2.46]*** | 0.94 [0.71–1.26] |
| **Education Level (In years)** | | |
| No education (**Ref**) | | |
| 1–7 | 0.57 [0.32–1.02]* | 0.89 [0.6–1.33] |
| 8–9 | 0.94 [0.52–1.7] | 1.25 [0.82–1.89] |
| 10 & above | 0.59 [0.33–1.06]* | 0.99 [0.62–1.6] |
| **Place of Residence** | | |
| Urban (**Ref**) | | |
| Rural | 0.90 [0.66–1.22] | 0.97 [0.69–1.37] |
| **Caste** | | |
| SC/ST (**Ref**) | | |
| OBC | 0.59 [0.4–0.86]*** | 0.95 [0.7–1.28] |
| Others | 0.68 [0.44–1.04]* | 0.92 [0.52–1.63] |
| **Religion** | | |
| Hindu (**Ref**) | | |
| Non-Hindu | 1.41 [1–1.98]** | 1.67 [1.08–2.58]** |
| **Wealth quintile** | | |

(*Continued*)

**Table 3.** (Continued)

| Background Characteristics | Unmarried | Married |
|---|---|---|
| | OR [95% CI] | OR [95% CI] |
| Poorest (**Ref**) | | |
| Poorer | 0.85 [0.49–1.48] | 1.14 [0.73–1.79] |
| Middle | 0.77 [0.43–1.39] | 1.04 [0.69–1.58] |
| Richer | 1.38 [0.8–2.38] | 0.81 [0.52–1.29] |
| Richest | 1.37 [0.73–2.58] | 0.97 [0.54–1.72] |
| **Mobile Access** | | |
| No Mobile (**Ref**) | | |
| Own Mobile | 1.46 [0.81–2.63] | 2.25 [1.18–4.27]** |
| Family member's | 1.00 [0.60–1.65] | 1.71 [0.88–3.3] |
| **Internet Access** | | |
| No (**Ref**) | | |
| Yes | 0.93 [0.57–1.52] | 1.95 [1.06–3.62]** |
| **Mass Media Exposure** | | |
| No (**Ref**) | | |
| Yes | 1.95 [1.30–0.87–1.96] | 0.95 [0.70–1.29] |
| **State** | | |
| Bihar (**Ref**) | | |
| Uttar Pradesh | 1.08 [0.8–1.45] | 1.54 [1.12–2.14]*** |

***p<0.01

**p<0.05

*p<0.10

Ref: Reference category; CI: Confidence interval; SC/ST: Scheduled caste/scheduled tribe; OBC: Other Backward Class; OR: Odds ratio; Violence among married women come under intimate partner violence; depressive symptoms (low include scores from 0–9 and high include scores 10–27).

odds to suffer from higher depressive symptoms compared to those who did not experience [OR: 1.98, 0.94–4.19].

Married girls who experienced emotional violence were 2.1 times more likely to have higher depressive symptoms than who do not face emotional violence [OR: 2.10, 1.42–3.11]. Moreover, married girls who had experienced physical violence had 63 per cent higher likelihood to suffer from higher depressive symptoms than their counterparts [OR: 1.63, 1.13–2.35]. Similarly, the likelihood of higher depressive symptoms was higher among adolescent girls who experienced sexual violence, though the results were not significant [OR: 1.11, 0.81–1.53].

Married girls who justified wife beating, had 59 per cent higher likelihood to suffer from higher depressive symptoms than those who did not justify [OR: 1.59, 1.09–2.33]. Unmarried girls who experienced harassment over mobile phone/internet had 78 per cent higher likelihood to suffer from higher depressive symptoms than who did not experience [OR: 1.78, 1.02–3.11]. Similarly, adolescents who perpetrated bullying had 90 per cent [OR: 1.90, 1.32–2.72] and 86 per cent [OR: 1.86, 0.98–3.52] higher likelihood to suffer from higher depressive symptoms among unmarried and married adolescents respectively compared to their counterparts. Married adolescents who had own mobile phone and internet access were 2.25 times [OR: 2.25, 1.18–4.27] and 1.95 times [OR: 1.95, 1.06–3.62] higher odds to suffer from higher depressive symptoms respectively than who did not have mobile phone access and no internet access.

Fig 1 displays result for interaction for depressive symptoms in married girls. The results present a scenario where the prevalence of depressive symptoms was categorized by

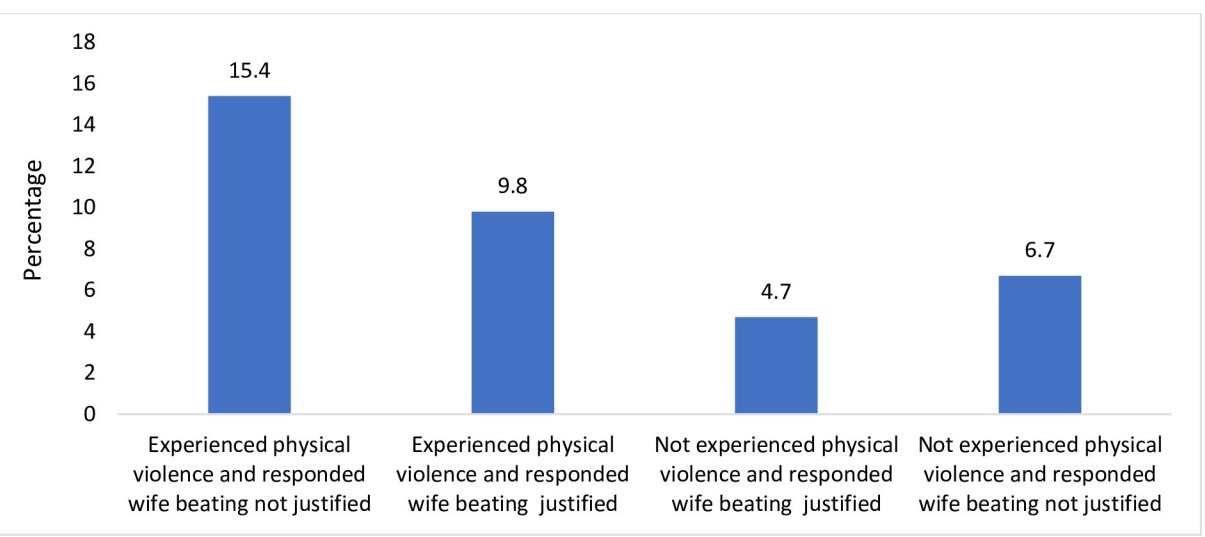

**Fig 1. Percentage of high depressive symptoms for physical violence and justification over wife beating among married adolescent girls.**

experiencing and not experiencing violence along with justifying wife beating. About 15 percent of married girls who experienced physical violence and did not justify wife beating had high depressive symptoms. Moreover, ten percent of married girls who experienced physical violence and justify wife beating had high depressive symptoms. The result clearly highlights that experiencing physical violence and not justifying wife beating were linked to higher depression among married adolescents.

Similarly, Fig 2 displays result for interaction for depressive symptoms in unmarried girls. The results present a scenario where the prevalence of depressive symptoms was categorized

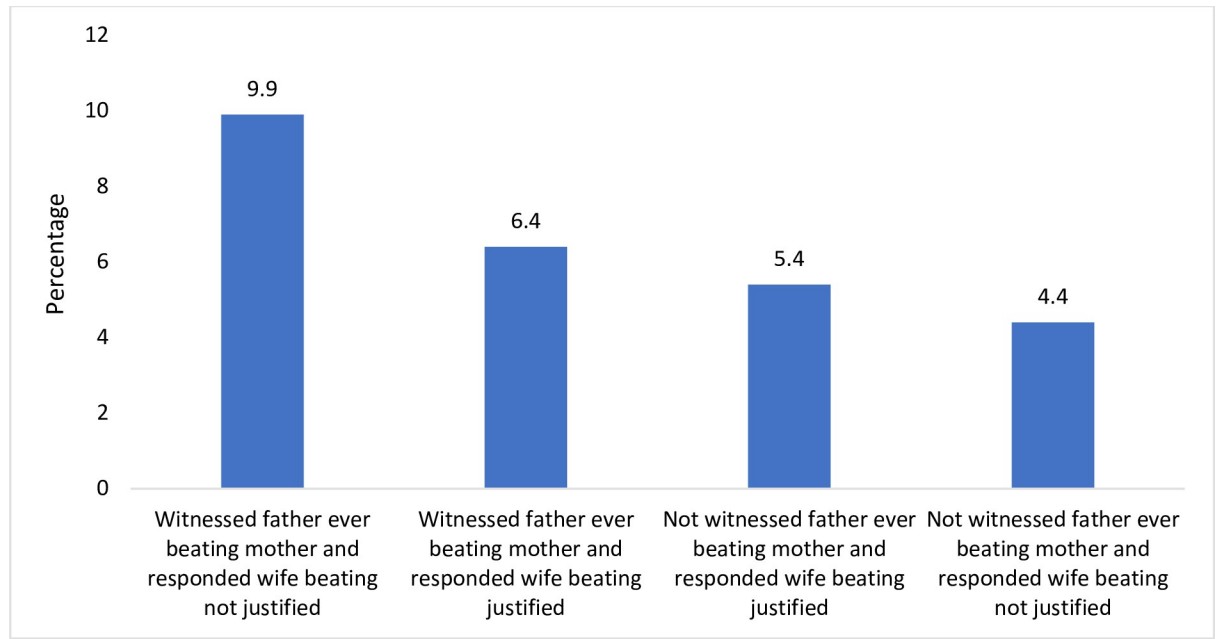

**Fig 2. Percentage of high depressive symptoms for unmarried adolescent girls who witnessed father ever beating mother and justification over wife beating.**

by witnessed and not witnessed violence against mother perpetuated by father along with justifying wife beating. Results found a higher prevalence of depressive symptoms for unmarried girls who witnessed father ever beating mother and did not justify wife beating. Moreover, six percent of unmarried girls who witnessed father ever beating mother and also justified wife beating had high depressive symptoms.

## Discussion

Previously, various studies have examined violence against women [29,30]; however, studies related to violence against married and unmarried adolescents are minimal across various settings [31–33]. Furthermore, minimal research is available examining the relationship between violence against adolescent girls and their mental health status [2,34]. By carrying out this study, we intended to fill the gap in the extant literature through exploring an association between violence against married and unmarried adolescents and depressive symptoms among them. Results noted that those who experience violence were more likely to face depressive symptoms. Previous studies also acknowledged that violence against women leads to mental health issues among them [35–38]. When in an abusive relationship, where women suffer violence against them, women lose self-confidence and therefore is more likely to face mental health issues [39].

Furthermore, women who suffer violence do not disclose this onslaught to anyone due to cultural secrecy and keep enduring the pain of violence until it becomes a mental trauma for them [40]. Results further found that adolescent girls who witness that their father beat mother were more likely to report depressive symptoms. It could also be important to understand in the context of this study that unmarried girls who witness IPV against mothers may also end up experiencing violence against them after marriage. The above notion has been confirmed through a few studies where evidence indicated that exposure to intimate partner violence against mother was one of the most common factors associated with male perpetration and female experiencing violence in later life [41,42].

Another important finding revealed that unmarried adolescent girls who experience internet based harassment were more likely to suffer from depressive symptoms. Studies have explicitly noticed that a subsequent number of adolescent girls face cyberbullying and online harassment [43,44]. Furthermore, studies also noticed mental health problems among adolescent girls who face cyberbullying and internet-based harassment [45,46]. The promotion of social networking sites such as Facebook, Instagram, and Snapchat has led to technology-facilitated harassment for girls [47,48]. Experiencing internet-based harassment may be an onset point for substance use, which may further perpetuate mental health issues among girls [49]. Perception over wife-beating is another crucial variable in the context of this study. This variable captures information about wife justifying violence against them. The study noticed that married adolescent girls who did not justify violence against them were more likely to show depressive symptoms. Previously studies have noted that women justify violence against them [50,51]. Deviating from the finding of this study, a study noticed that women who believe that intimate partner violence is acceptable were found to be having long-term mental health problems [52]. However, the reverse finding from this study could be attributed to the fact that married adolescent girls who do not justify violence against them may be experiencing severe violence from their husbands which may be aggravating mental trauma among them leading to higher depressive symptoms.

Several background characteristics also explain the onset of depressive symptoms as a result of violence against adolescents. Education has been observed as a protective factor against depressive symptoms among unmarried adolescent girls who have faced violence being

perpetrated. Previously, various studies have noted the importance of education in lowering the onset of violence [53–55]; however, studies examining the importance of education in lowering the violence and subsequently mental health issues among adolescent girls are minimal [56]. Increasing education levels among women improve their chances of personal skills and employability, which further decreases their risk of being exposed to violence [57]. Furthermore, parental education could also averse the violence against women [58]. Caste is another important variable in the context of this study. Results noted that unmarried girls from the General caste category were less likely to show depressive symptoms than their counterparts. Population from the General caste category belong to the well-off socio-economic group as compared to the population in other remaining caste categories such as Scheduled Caste, Scheduled Tribe, and Other Backward Castes; therefore, women in marginalized caste experience higher intimate partner violence [59]. Furthermore, women in Scheduled caste experience higher violence as they belong to the economically weaker section and are less likely to be employed [60]. The higher rates of violence among the marginalized group could be attributed to their worsened mental health and subsequently higher depressive symptons.

The study has several limitations. There is a high possibility of under-reporting of information on violence. Previous studies have also noted that it is not easy to extract information on violence against women due to various socio-cultural issues [61,62]. Furthermore, information on depressive symptoms was self-reported and not clinically diagnosed. Also, cross-sectional nature of data limits our understanding of causal inference. At last, the study was based on two states of India, namely, Uttar Pradesh and Bihar, and therefore study findings shall not be generalized to the national population. Despite the above limitations, the study provides first-hand information on the relationship between violence and depressive symptoms among unmarried and married adolescent girls.

## Conclusion

We believe that our study is among the first of its kind in examining the association between violence and depressive symptoms among married and unmarried adolescent girls in two backward states in India. The strength of the study is its finding extend beyond intimate partner violence upon married women and included various covariates that explained violence and depressive symptoms among unmarried adolescent girls. We identified several significant findings from this study. Married and unmarried adolescents who faced any form of violence were more likely to experience depressive symptoms. Furthermore, internet-based harassment and, subsequently, depressive symptoms were also noticed among the study population. The findings from this study call out the need to implement policies that explore to recognize and battle depression or its onset among adolescents and also bring the issue of violence against women to the forefront. It is recommended that the state governments of Uttar Pradesh and Bihar shall oblige to prevent violence against women, and further state-governments should provide funding for research and services that uptake the violence against women to the core issue. It is important to protect the health and well-being of women, and therefore, it is recommended that state governments shall ensure that victims of any form of violence must have access to health-care services.

## Acknowledgments

### Ethical statement

Informed consent was sought from each individual to be interviewed, among unmarried adolescents in ages 10–17, consent was also sought from a parent or guardian. Additionally,

names were never recorded in the computer form in which data were collected. In order to preserve the confidentiality of the respondent or the parent/guardian, signing the consent form was optional; however, the interviewer was required to sign a statement that she or he had explained the content of the consent form to the respondent or parent.

## Author Contributions

**Conceptualization:** Ratna Patel, Shobhit Srivastava, Pradeep Kumar, Shekhar Chauhan, Mani Deep Govindu.

**Data curation:** Shobhit Srivastava, Pradeep Kumar.

**Formal analysis:** Shobhit Srivastava, Pradeep Kumar, Mani Deep Govindu.

**Investigation:** Shobhit Srivastava, Pradeep Kumar.

**Methodology:** Shobhit Srivastava, Mani Deep Govindu.

**Software:** Shobhit Srivastava, Pradeep Kumar, Mani Deep Govindu.

**Supervision:** Preeti Dhillon.

**Validation:** Shobhit Srivastava, Pradeep Kumar, Mani Deep Govindu, Preeti Dhillon.

**Writing – original draft:** Ratna Patel, Samriddhi S. Gupte, Shekhar Chauhan.

**Writing – review & editing:** Ratna Patel, Samriddhi S. Gupte, Shekhar Chauhan.

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
