## [Decision Letter · Decision Letter 0]

30 Dec 2020

PONE-D-20-32760

Experience of Gender-based Violence and its effect on the mental health status of Indian adolescent girls: Evidence from UDAYA survey

PLOS ONE

Dear Dr. Chauhan,

Thank you for submitting your manuscript to PLOS ONE. After careful consideration, we feel that it has merit but does not fully meet PLOS ONE’s publication criteria as it currently stands. Therefore, we invite you to submit a revised version of the manuscript that addresses the points raised during the review process.

We look forward to receiving your revised manuscript.

Kind regards,

Kannan Navaneetham, PhD

Academic Editor

PLOS ONE

2. We note that you have reported significance probabilities of 0 in places. Since p=0 is not strictly possible, please correct this to a more appropriate limit, eg 'p<0.0001'.

Furthermore, within the Methods section, please provide further clarification regarding how the wealth quintile was stratified for analysis.

3. Please ensure you have also stated whether consent was obtained from parents or guardians of the minors included in the study, and how this was documented. Or whether the research ethics committee or IRB specifically waived the need for their consent.

Reviewers' comments:

Reviewer's Responses to Questions

**Comments to the Author**

1. Is the manuscript technically sound, and do the data support the conclusions?

Reviewer #1: Yes

Reviewer #2: Partly

2. Has the statistical analysis been performed appropriately and rigorously? 

Reviewer #1: Yes

Reviewer #2: No

3. Have the authors made all data underlying the findings in their manuscript fully available?

Reviewer #1: Yes

Reviewer #2: Yes

4. Is the manuscript presented in an intelligible fashion and written in standard English?

Reviewer #1: No

Reviewer #2: No

5. Review Comments to the Author

Reviewer #1: This paper tackles an important subject the mental health consequences of exposure to violence in adolescent girls and women in India.

Abstract, Conclusion: This is not the first study worldwide to associate violence and mental health in adolescent girls. Please modify this statement. Is it the first study in India?

Page 3, line 4: ‘understanding of violence of include’ You don’t need the second ‘of’ here.

Page 4, line 2: I don’t understand this phrase ‘per lakh’.

Page 4, line 8: Can you clarify this ‘Given the diversity and disparity that exists’.

Page 4, line 4 up: This is not true. There are studies of violence and its mental health consequences in terms of depressive symptoms, anxiety symptoms and PTSD in the published literature.

Page 5, first paragraph: Can you please state your hypotheses here.

Page 5, line 15: How was the sample analysed in this study derived from the population of 20,000 participants in the original study? Was it a representative sample?

Page 5, line 16: What is a ‘Gauna’? Please add an explanation for those readers not familiar with Indian culture.

Page 6, Outcome variable: Was this measure of depressive symptoms a standard scale – if so, please reference it? If it was not a standard scale where was it derived from? How did you decide on the threshold for depressive symptoms?

Page 7, line 7: Did you measure whether they were a victim of bullying as well as carrying out bullying?

Page 8, Results: I wonder if it would be better if you began your results with more general findings, rather than the interactions i.e. Table 1. This would help to set the context for the interaction results. It is not clear why you stratify your results by whether it is justifiable for husbands to beat their wives?

Table 1: What does ‘OBC Other backward Class’ mean? A more detailed explanation would help readers not familiar with Indian Government Surveys.

Page 17, line 3: There are more studies of violence and mental health in adolescents than you report. It would be worth you doing further literature searches. This is not to undermine the importance of your findings.

Page 18, paragraph 2: Could the protective effect of education against violence in unmarried girls be a reflection of the protective effect of parental education levels?

In 'Limitations' you could also mention that as this is a cross sectional study there are limits to how much you can justify causal statements in the paper.

In general the English grammar in the paper could be improved before resubmission.

Reviewer #2: Dear Autors,

Your manuscript seems to be very interesting. You expose the problem with experiencing violence by young girls in India, which is rare. The manuscript requires many changes in the content or should be submitted to a different journal.

Abstract: it should specify in more detail the aim of the study. In my opinion, there is some discrepancy between the abstract and the main body of the manuscript in terms of aim of the study.

Some sentences cited information require bibliographic confirmation (eg, on p. 3, “Global studies show women all over the world face violence in various forms at hands of various people.”).

The authors do not specify the outcome variable. It first appears as an outcome variable, mental health, and then depressive symptoms. It is not clear what the depressive symptoms mean. It is not known what the definition of an outcome variable is and how this variable is described. The authors do not refer to the universal DSM 5 classification system or even any other.

Detailed information on how the statistics are calculated should not appear in the manuscript, as shown on p. 14.

The study aimed to test the impact of violence on the mental health of adolescents' victims. Unfortunately, the statistics do not allow to measure the value of impact or rather the prediction the odds of being a case based on the values of the independent variables (predictors).

According to Guidelines for reviewers, your manuscript should be revisted.

6. PLOS authors have the option to publish the peer review history of their article (what does this mean?). If published, this will include your full peer review and any attached files.

Reviewer #1: **Yes: **Stephen Stansfeld

Reviewer #2: No

---

## [Author Response · Author response to Decision Letter 0]

21 Jan 2021

Editor’s Comments:

 Response: The manuscript has been prepared as per the journal’s guidelines.

2. We note that you have reported significance probabilities of 0 in places. Since p=0 is not strictly possible, please correct this to a more appropriate limit, eg 'p<0.0001'.

Furthermore, within the Methods section, please provide further clarification regarding how the wealth quintile was stratified for analysis.

Response: ***p<0.001, **p<0.05, *p<0.10 is now used in the manuscript. Also, clarification regarding how the wealth quintile was created is also added. 

 3. Please ensure you have also stated whether consent was obtained from parents or guardians of the minors included in the study, and how this was documented. Or whether the research ethics committee or IRB specifically waived the need for their consent.

Response: The data used is secondary data and therefore authors did not require any ethical consent. However, the agencies that collected the data undertook followed all the required ethical guidelines. Please refer to the following references for the same. 

1. Santhya KG, Acharya R, Pandey N, Gupta A, Rampal S, Singh S, Zavier AF. Understanding the lives of adolescents and young adults (UDAYA) in Uttar Pradesh, India. New Delhi. 2017. https://www.popcouncil.org/uploads/pdfs/2017PGY_UDAYA-UPreport.pdf

2. Santhya KG, Acharya R, Pandey N, Singh SK, Rampal S, Zavier AJ, Gupta AK. Understanding the lives of adolescents and young adults (UDAYA) in Bihar, India. https://www.popcouncil.org/uploads/pdfs/2017PGY_UDAYA-BiharReport.pdf

Reviewer #1: 

This paper tackles an important subject the mental health consequences of exposure to violence in adolescent girls and women in India.

1. Abstract, Conclusion: This is not the first study worldwide to associate violence and mental health in adolescent girls. Please modify this statement. Is it the first study in India?

Response: We have modified the statement as suggested.

2. Page 3, line 4: ‘understanding of violence of include’ You don’t need the second ‘of’ here.

Response: Authors are thankful to the reviewer for pointing out the mistake. Accordingly changes have been made.

3. Page 4, line 2: I don’t understand this phrase ‘per lakh’.

Response: Lakh is the one-tenth part of the ‘million.’ One million equals to 10 lakhs. However for the general understanding, we have changed it to million in the text.

4. Page 4, line 8: Can you clarify this ‘Given the diversity and disparity that exists’.

Response: It means that India is a country with wide diversity and disparity. India has around 29 states. States have their own language and culture varying from each other.

5. Page 4, line 4 up: This is not true. There are studies of violence and its mental health consequences in terms of depressive symptoms, anxiety symptoms and PTSD in the published literature.

Response: Changes have been made as suggested.

6. Page 5, first paragraph: Can you please state your hypotheses here.

Response: Authors have stated hypothesis as suggested by the reviewer.

7. Page 5, line 15: How was the sample analysed in this study derived from the population of 20,000 participants in the original study? Was it a representative sample?

Response: Yes, the sample was representative. Additionally, only sample from married and unmarried adolescent girls were taken for the analysis. The girls whose ‘gauna’ was not performed were not added in the analysis. The term ‘gauna’ has been defined in the manuscript.

8. Page 5, line 16: What is a ‘Gauna’? Please add an explanation for those readers not familiar with Indian culture.

Response: Authors have defined the term ‘gauna’ in the revised manuscript.

9. Page 6, Outcome variable: Was this measure of depressive symptoms a standard scale – if so, please reference it? If it was not a standard scale where was it derived from? How did you decide on the threshold for depressive symptoms?

Response: Yes, the scale was a standard scale. The reference is now added as per given suggestion. 

10. Page 7, line 7: Did you measure whether they were a victim of bullying as well as carrying out bullying?

Response: The respondents were carrying out bullying. The change is incorporated in the manuscript.

11. Page 8, Results: I wonder if it would be better if you began your results with more general findings, rather than the interactions i.e. Table 1. This would help to set the context for the interaction results. It is not clear why you stratify your results by whether it is justifiable for husbands to beat their wives?

Response: Authors are thankful to the reviewer for pointing out the mistakes. Accordingly, changes have been made as suggested by the reviewer. We stratified our results by whether it is justifiable for husbands to beat their wives because it is expected that those women who justify wife beating may find it normal and therefore may not report depressive symptoms due to violence. The explanation has been given the manuscript.

12. Table 1: What does ‘OBC Other backward Class’ mean? A more detailed explanation would help readers not familiar with Indian Government Surveys.

Response: Details are now added as per suggestion. The information regarding OBC and other caste group has been added to exposure variable section.

13. Page 17, line 3: There are more studies of violence and mental health in adolescents than you report. It would be worth you doing further literature searches. This is not to undermine the importance of your findings.

Response: As per given suggestion, authors have included other relevant studies to their manuscript.

14. Page 18, paragraph 2: Could the protective effect of education against violence in unmarried girls be a reflection of the protective effect of parental education levels?

Response: Indeed parental education acts as a safety net. Previous research also outlined the protective effect of parental education in reducing the likelihood of violence (Yakubovich, A. R., Stöckl, H., Murray, J., Melendez-Torres, G. J., Steinert, J. I., Glavin, C. E., & Humphreys, D. K. (2018). Risk and protective factors for intimate partner violence against women: Systematic review and meta-analyses of prospective–longitudinal studies. American journal of public health, 108(7), e1-e11.).

15. In 'Limitations' you could also mention that as this is a cross sectional study there are limits to how much you can justify causal statements in the paper.

Response: The authors are thankful to the reviewer for pointing out the genuine limitation. Accordingly, we have included the limitation as suggested.

16. In general the English grammar in the paper could be improved before resubmission.

Response: The authors have taken help from one independent native English speaker to improve the English grammar of the paper.

Reviewer #2: 

1. Your manuscript seems to be very interesting. You expose the problem with experiencing violence by young girls in India, which is rare. The manuscript requires many changes in the content or should be submitted to a different journal.

Response: Authors are thankful to the reviewer for acknowledging the importance of this study. Also, authors have modified the manuscript as suggested by two independent reviewers.

2. Abstract: it should specify in more detail the aim of the study. In my opinion, there is some discrepancy between the abstract and the main body of the manuscript in terms of aim of the study.

Response: the abstract has been modified as suggested by the reviewer. The background of the study has been modified and now the aim is clear as discussed in the manuscript.

3. Some sentences cited information require bibliographic confirmation (eg, on p. 3, “Global studies show women all over the world face violence in various forms at hands of various people.”).

Response: Authors have provided the citation at the required place in the text as suggested by the reviewer. 

4. The authors do not specify the outcome variable. It first appears as an outcome variable, mental health, and then depressive symptoms. It is not clear what the depressive symptoms mean. It is not known what the definition of an outcome variable is and how this variable is described.

Response: The outcome variable was depressive symptoms coded as low and high. The reference is added. Outcome variable is generally known as dependent variable. Outcome variables are observed and measured by changing independent variables. These variables determine the effect of the cause (independent) variables when changed for different values.

5. The authors do not refer to the universal DSM 5 classification system or even any other.

Response: Authors did not refer to DSM 5 as this study referred PHQ-9 to measure depressive symptoms. Proper citation has also been provide for the same.

6. Detailed information on how the statistics are calculated should not appear in the manuscript, as shown on p. 14.

Response: Comment incorporated. 

7. The study aimed to test the impact of violence on the mental health of adolescents' victims. Unfortunately, the statistics do not allow to measure the value of impact or rather the prediction the odds of being a case based on the values of the independent variables (predictors).

Response: Authors are thankful to the reviewer for pointing out the mistake. Actually, authors did not intend to test the impact. It was an error while writing the manuscript. Authors intend to examine the association between violence and depressive symptoms among married and unmarried adolescents. Accordingly, we have replaced the word ‘impact’ with the word ‘association’ throughout the manuscript.

8. According to Guidelines for reviewers, your manuscript should be revisited.

Response: Authors have modified the manuscript as per given suggestions. Authors have strictly followed the guidelines while preparing the revised version of the manuscript.

---

## [Decision Letter · Decision Letter 1]

26 Feb 2021

Experience of Gender-based Violence and its effect on depressive symptoms among Indian adolescent girls: Evidence from UDAYA survey

PONE-D-20-32760R1

Dear Dr. Chauhan,

We’re pleased to inform you that your manuscript has been judged scientifically suitable for publication and will be formally accepted for publication once it meets all outstanding technical requirements.

Kind regards,

Kannan Navaneetham, PhD

Academic Editor

PLOS ONE

Additional Editor Comments (optional):

Reviewers' comments:

Reviewer's Responses to Questions

**Comments to the Author**

1. If the authors have adequately addressed your comments raised in a previous round of review and you feel that this manuscript is now acceptable for publication, you may indicate that here to bypass the “Comments to the Author” section, enter your conflict of interest statement in the “Confidential to Editor” section, and submit your "Accept" recommendation.

Reviewer #1: All comments have been addressed

Reviewer #2: All comments have been addressed

2. Is the manuscript technically sound, and do the data support the conclusions?

Reviewer #1: Yes

Reviewer #2: Yes

3. Has the statistical analysis been performed appropriately and rigorously? 

Reviewer #1: Yes

Reviewer #2: Yes

4. Have the authors made all data underlying the findings in their manuscript fully available?

Reviewer #1: Yes

Reviewer #2: Yes

5. Is the manuscript presented in an intelligible fashion and written in standard English?

Reviewer #1: Yes

Reviewer #2: Yes

6. Review Comments to the Author

Reviewer #1: (No Response)

Reviewer #2: As I mentioned in the first review, the research issue is very interesting and important. You made changes according to my comments. The theoretical background seems to be improved.

The most important concern of your manuscript is showing the phenomenon of violence taking into account the cultural context.

7. PLOS authors have the option to publish the peer review history of their article (what does this mean?). If published, this will include your full peer review and any attached files.

Reviewer #1: **Yes: **Stephen Stansfeld

Reviewer #2: **Yes: **Magdalena Rode

---

## [Editor Report · Acceptance letter]

16 Mar 2021

PONE-D-20-32760R1 

Experience of Gender-based Violence and its effect on depressive symptoms among Indian adolescent girls: Evidence from UDAYA survey  

Dear Dr. Chauhan:

I'm pleased to inform you that your manuscript has been deemed suitable for publication in PLOS ONE. Congratulations! Your manuscript is now with our production department. 

Kind regards, 

on behalf of

Professor Kannan Navaneetham 

Academic Editor

PLOS ONE